# Light-triggered release of conventional local anesthetics from a macromolecular prodrug for on-demand local anesthesia

Wei Zhang [1], Tianjiao Ji[1], Yang Li[1], Yueqin Zheng[1], Manisha Mehta[1], Chao Zhao[1], Andong Liu[1] & Daniel S. Kohane [1✉]

An on-demand anesthetic that would only take effect when needed and where the intensity of anesthesia could be easily adjustable according to patients' needs would be highly desirable. Here, we design and synthesize a macromolecular prodrug (P407-CM-T) in which the local anesthetic tetracaine (T) is attached to the polymer poloxamer 407 (P407) via a photo-cleavable coumarin linkage (CM). P407-CM-T solution is an injectable liquid at room temperature and gels near body temperature. The macromolecular prodrug has no anesthetic effect itself unless irradiated with a low-power blue light emitting diode (LED), resulting in local anesthesia. By adjusting the intensity and duration of irradiation, the anesthetic effect can be modulated. Local anesthesia can be repeatedly triggered.

---

[1] Laboratory for Biomaterials and Drug Delivery, Department of Anesthesiology, Division of Critical Care Medicine, Boston Children's Hospital, Harvard Medical School, Boston, MA, USA. ✉email: daniel.kohane@childrens.harvard.edu

 1

ocalized pain can be treated by systemic or local medications. The former often include opioids, which have many associated problems and side effects, including nausea, itching, constipation, tolerance, addiction, death by overdose, and the potential for diversion. Local treatments commonly involve local anesthetics. These are very effective but are of relatively brief duration. Moreover, their initial administration is painful (if done in awake patients), and requires skilled personnel in a medical facility. Consequently, there has been interest, for many decades, in developing sustained release systems that would provide prolonged duration local anesthesia[1–3]. However, those systems released drug in a relatively monotonic manner, and did not have a mechanism by which the drug release—and the resulting local anesthesia—could be modulated in real time in response to changing patient needs. A local anesthetic release system which could be easily adjusted by patients could be of benefit.

To achieve that end, we have employed drug-delivery systems that are responsive to external stimuli, in which a therapeutic effect can be achieved at a desired dosage, time, and location, ideally repeatedly[4–6]. A variety of triggers have been used in the literature, such as light[7–10], ultrasound[11,12] and magnetic fields[13,14]. Among them, light is a promising trigger because of its tunable wavelength, irradiance, and area and duration of exposure[15]. We have developed liposomal systems in which local anesthesia was achieved by irradiation of liposomal carriers, such that the time of onset, intensity, and duration of local anesthesia could be determine by the timing, intensity, and duration of irradiation[16–18]. Previously triggered local anesthetic sustained release systems suffered from a problem common to most particulate drug-delivery systems: release occurring from the moment of the devices' creation until drug is depleted. Early on, this results in untriggered rapid drug release; in the context of local anesthesia, this may result in extended initial nerve block, which may be undesirable. Subsequently, ongoing release may result in depletion of drug even if the system is not triggered (i.e., basal release), so that it is no longer available for triggered release. To eliminate the unwanted initial blockade and basal drug release, we conjugated the drug onto macromolecular carriers in a manner that could be reversed by photo-triggering.

We have designed a light-triggerable polymer–drug conjugate, which is composed of three parts: a conventional local anesthetic, a polymer carrier, and a photo-cleavable linkage in between them (Fig. 1a). Conventional local anesthetics usually consist of a hydrophobic aromatic ring, an intermediate linkage (ester or amide) and a tertiary amine. Some of them have a primary amine (as in procaine and chloroprocaine) or secondary amine (as in tetracaine) substituent group on the aromatic ring, which could be used in chemical reactions. Tetracaine is selected as the local anesthetic in this work owing to its widespread clinical use, relatively high potency in its class and the presence of a modifiable secondary amine group[19]. Poloxamer 407 (P407), which is a Food and Drug Administration-approved polymer, is used as the polymeric carrier. Apart from acting as the macromolecular component of the prodrug, it has the desirable property of reverse thermal gelation, i.e., a solution of P407 is liquid at low (e.g., room) temperature during injection and gels in the body owing to the higher temperature[20]. This gelation encourages persistence of the polymer at the site of injection. 7-(diethylamino) coumarin (DEACM) is selected as the photo-responsive moiety owing to its good photo-cleavage efficiency, relatively long absorption wavelength (~400 nm, blue light, low phototoxicity) as well as good stability in darkness, which may be beneficial for shelf-life[15,21–23].

We hypothesized that this system would enable light-triggerable nerve block over an extended period, with minimal untriggered drug release. Consequently, there would be no initial nerve block and no wastage of drug between triggered events. By tuning the intensity and duration of irradiation, the amount of drug released could be adjusted on-demand, allowing modulation of the anesthetic effect. We use a light-emitting diode (LED) as the light source, because LEDs are cheaper and easier to use than lasers. Moreover, the lower energies involved suggest a lower risk of thermal injury.

## Results

**Synthesis and characterization of P407-CM-T.** The molecular design and synthesis of the polymer–drug conjugate (P407-couramin-tetracaine; P407-CM-T) are shown in Fig. 1a. The photo-cleavage reaction is shown in Fig. 1b. The detailed synthetic procedures, additional discussion, and yield of each step are in Supplementary Methods. In brief, a coumarin derivative bi-functionalized at the 4-position of the coumarin ring was synthesized (see Supplementary Fig. 1), then an azide group was introduced for click reaction with the P407. Then tetracaine was connected to the coumarin through a carbamate linkage. The resulting coumarin-tetracaine conjugate was then conjugated onto an alkyl-functionalized P407 through azide-alkyne click cycloaddition[24,25]. After the reaction, the molecular weight of the polymer (determined with a polystyrene standard) as measured by gel permeation chromatography (GPC) increased from 12.6k to 13.6k. In the $^1$H NMR spectrum of P407-CM-T (Fig. 1c), peaks could be clearly assigned to each moiety of the molecule, indicating its successful synthesis: the peak at 8.03–8.04 ppm is representative of the tetracaine moiety; the peaks at 6.70–6.68 ppm, 6.49–6.48 ppm, and 6.31–6.29 ppm are representative of the coumarin moiety[26]; and the large peaks at 3–4 ppm and 1–2 ppm are attributable to P407 by comparing to the spectrum of alkyne terminated P407 in Supplementary Fig. 9[27]. These results demonstrated the successful synthesis of the designed material. More detailed chemical characterizations of all compounds are available in Supplementary Figs. 4–12.

P407-CM-T in aqueous solution showed strong absorption in the blue range with a maximum at 394 nm and a full width at half-maximum of 64 nm (Fig. 2a). By comparing to the absorption spectra of coumarin and tetracaine, the absorption peak of P407-CM-T at ~400 nm can be attributable to the coumarin moiety, and the peak at ~300 nm to the tetracaine moiety. DEACM derivatives are known to be cleavable by 400 nm LED[26,28]. Therefore, a 400 nm LED with tunable intensity was used in the triggering experiments.

The reverse thermal gelation property of P407-CM-T (20 wt% in aqueous) was verified by oscillatory shear rheology (10 rads$^{-1}$, 1% strain, 1 °C min$^{-1}$). The loss modulus (G″) was higher than the storage modulus (G′) at <30 °C, indicating a liquid-like property, which ensured the material would be injectable. G′ became greater than G″ at ~33 °C, indicating gelation (Fig. 2b). This property could be beneficial for the material to be retained at the site of injection[29], where drug release could subsequently be triggered.

The ability of P407-CM-T to release tetracaine under 400 nm light was studied in vitro (Fig. 3a). Irradiation of P407-CM-T (10 mg mL$^{-1}$) with 400 nm light at 50 mW cm$^{-2}$ revealed a peak on liquid chromatography that was not seen in unirradiated P407-CM-T and that co-migrated with free tetracaine. The small molecule released from P407-CM-T after irradiation had the molecular weight of tetracaine ion ($m/z$ of 265.1) by liquid chromatography–mass spectroscopy (LC-MS) (Fig. 3b), indicating that P407-CM-T can release tetracaine in its native form. Almost all the conjugated drug was cleaved from P407-CM-T (10 mg mL$^{-1}$) within 120 s of irradiation at 50 mW cm$^{-2}$, and ~40% of cleavage occurred within 120 s at 25 mW cm$^{-2}$ (Fig. 3c). When P407-CM-T solution was kept in the dark at 37 °C for

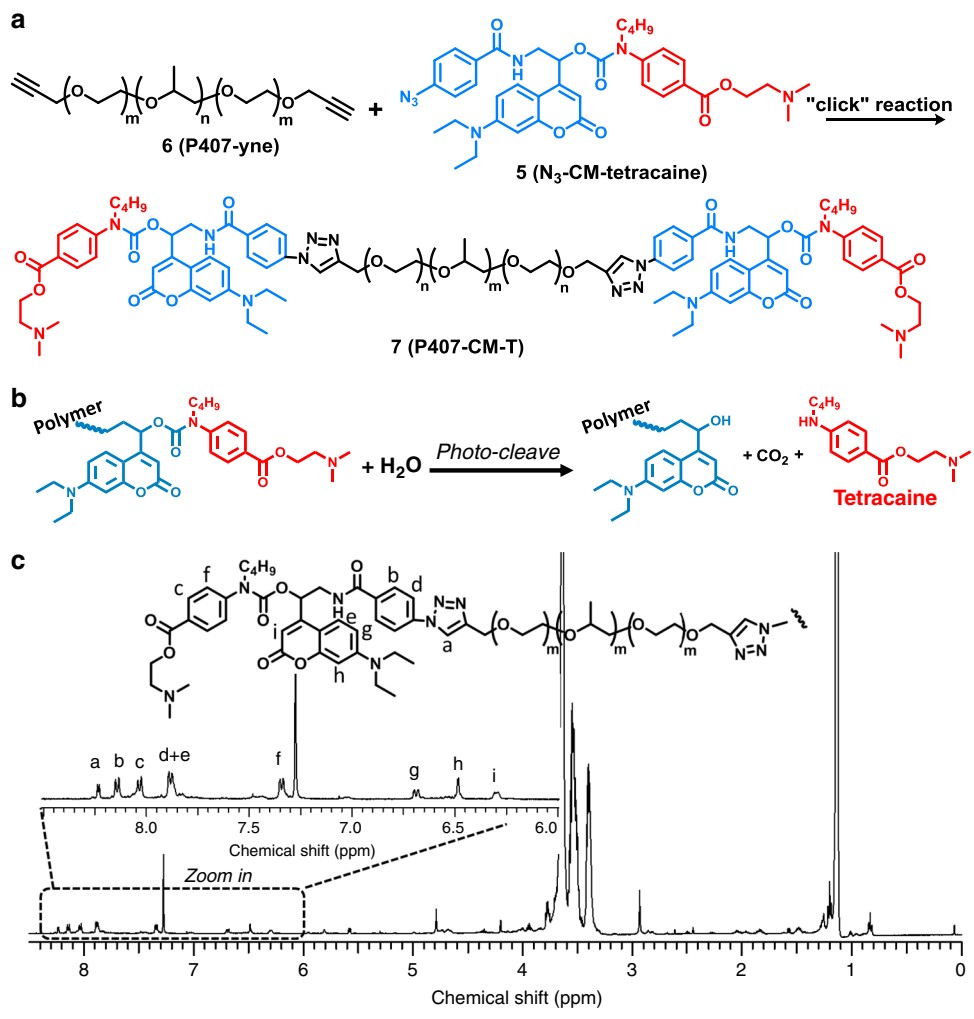

**Fig. 1 The light-triggerable tetracaine-polymer prodrug. a.** Synthesis of P407-CM-T through alkyne-azide click reaction. **b** Photo-cleavage reaction of P407-CM-T. **c** ¹H NMR of P407-CM-T.

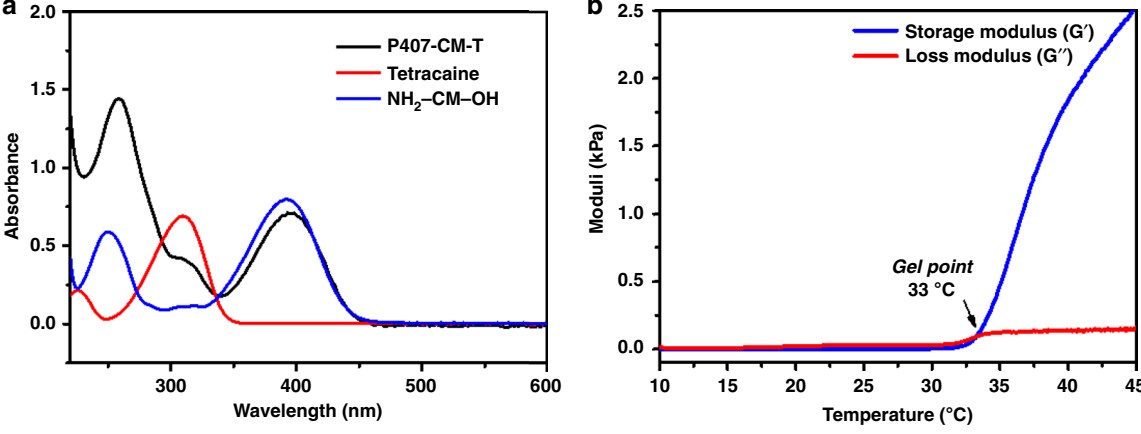

**Fig. 2 Physical properties of P-CM-T. a** UV-vis absorption spectra of aqueous solutions of P-CM-T (0.3 mg mL⁻¹, black line), tetracaine hydrochloride (0.01 mg mL⁻¹, red line) and NH₂-CM-OH (0.01 mg mL⁻¹, blue line). **b** Rheology of 20 wt% P-CM-T as a function of temperature (blue line: G′; red line: G″).

2 weeks, no tetracaine was released (Supplementary Fig. 13), indicating good stability in the dark.

**In vitro biocompatibility.** Before testing in vivo, materials were evaluated in vitro with C2C12 and PC12 cells to assess

cytotoxicity to muscle and nerve cells, respectively. Different compounds were directly added into the cell culture media (1 mg mL⁻¹, 0.33 mg mL⁻¹, and 0.11 mg mL⁻¹) and incubated in the media bathing the cells (i.e., in direct contact with them) in conventional cell culture wells. After 24 h, cell viabilities were evaluated with the MTS assay, and their survival expressed as

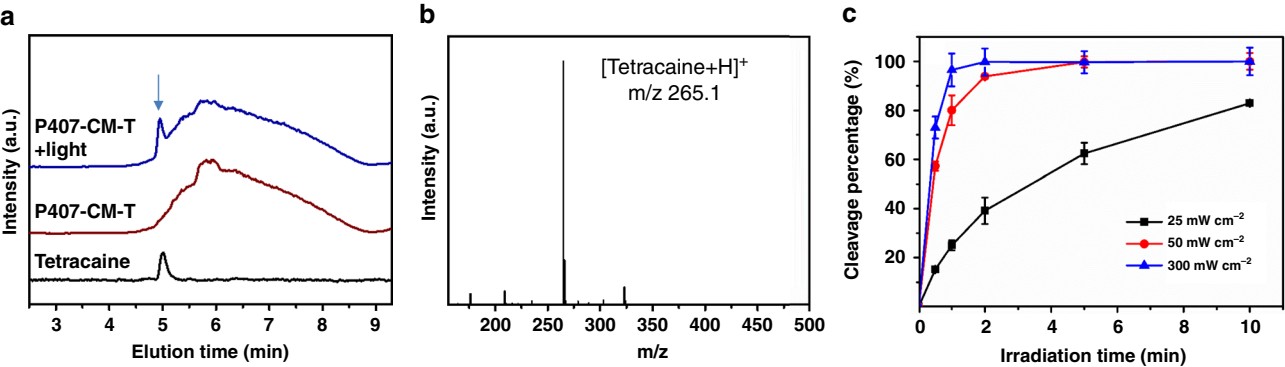

**Fig. 3 In vitro photocleavage of P407-CM-T. a** Liquid chromatography curves of tetracaine (black line), P407-CM-T before (brown line) and after cleavage (blue line) under 400 nm LED irradiation. **b** Mass spectrum corresponding to the peak at 5 min in the curve of P407-CM-T + light in **a**. **c** Photocleavage of P407-CM-T solution (10 mg mL$^{-1}$) over time under 400 nm LED at different irradiances ($n = 4$, Data are means ± SD) (black line: 25 mW cm$^{-2}$; red line: 50 mW cm$^{-2}$; blue line: 300 mW cm$^{-2}$). Source data are provided as a Source Data file.

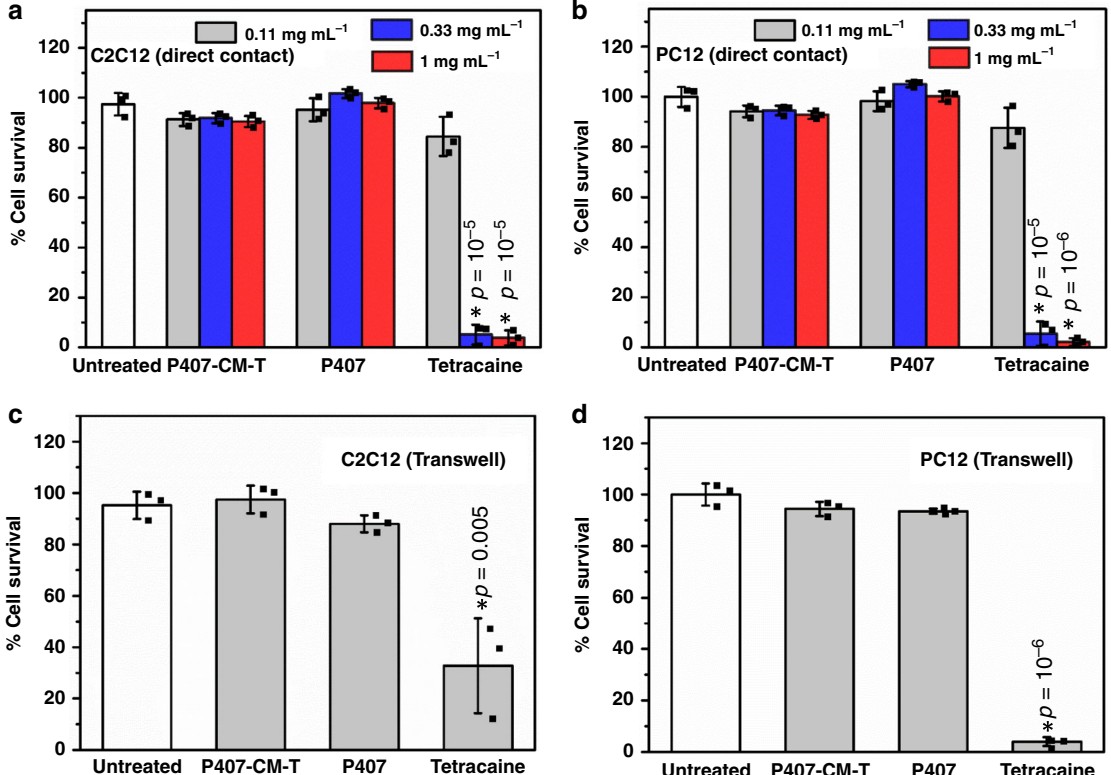

**Fig. 4 Cytotoxicity of P407-CM-T. a**, **b**. Cell survival (determined by MTS assay) of C2C12 and PC12 cells in direct contact with various concentrations of tetracaine, P407 and P407-CM-T for 24 h (gray: 0.11 mg mL$^{-1}$; blue: 0.33 mg mL$^{-1}$; red: 1 mg mL$^{-1}$). **c**, **d** Cell survival (determined by MTS assay) of C2C12 and PC12 cells exposed to tetracaine (0.5% wt%), P407 (20% wt%) and P407-CM-T (20% wt%) in Transwells for 24 h ($n = 3$).* indicates $p < 0.05$ (unpaired $t$ test) compared with untreated cells. Data are means ± SD. Source data are provided as a Source Data file.

percentages of results in untreated cells. Free tetracaine showed high cytotoxicity in both cell lines, while P407 and P407-CM-T showed little cytotoxicity (Fig. 4a, b). Owing to the high molecular weight of the polymer-containing samples, their molar concentrations were perforce lower than that of the small molecule tetracaine. It would not be practical to culture cells in media containing the polymer concentration to be used in vivo (20 wt %), which would be very viscous. Therefore, test materials were placed in Transwell inserts such that they were in continuity with the cell culture media. Here, also both P407 and P407-CM-T had no effect on cell viability ($p > 0.05$ compared with untreated cells) while tetracaine greatly reduced cell survival ($p < 0.05$ compared

with untreated cells) in both cell lines (Fig. 4c, d). These results indicate that covalently linking tetracaine onto P407 can effectively reduce its cytotoxicity, presumably because there was little free tetracaine.

**In vivo efficacy**. The effectiveness of light-triggered anesthesia using P407-CM-T was assessed in vivo after footpad injection in the rat (Table 1)[17,30–34]; this model was selected because it was anticipated that 400 nm light would not penetrate deeply into tissue[35]. A hundred microliters of 20 wt% P407-CM-T were injected into the plantar aspect of the rat hind paw.

Neurobehavioral testing was performed by stimulating the rat footpad with a Touch Test sensory evaluator and noting the vocal and/or motor response (foot withdrawal) of the rat. Maximum peak effect (MPE) and duration of block were calculated (see Methods)[17,30]. Injection of 100 µL of saline or P407-CM-T (20 wt %) did not affect response to the filament (Supplementary Fig. 14). In the absence of irradiation, P407-CM-T did not cause local anesthesia (Fig. 5a). Irradiation of the site of administration immediately after injection with a 400 nm LED at 200 mW cm$^{-2}$ for 2 minutes caused local anesthesia lasting 19.5 ± 4.5 min (Fig. 5a and b). (Higher irradiances were used in vivo compared with in vitro owing to attenuation of light by traversing tissues[36].) Local anesthesia was prolonged to 36.5 ± 8.9 min (Fig. 5b) by increasing the irradiance to 300 mW cm$^{-2}$ also for 2 minutes (p <

0.05 compared with 200 mW cm$^{-2}$). Extending the irradiation time to five minutes at 300 mW cm$^{-2}$ further extended local anesthesia to 66.7 ± 24 min (p < 0.05 compared with 300 mW cm$^{-2}$ for 2 min, Fig. 5b). Tetracaine conjugated to coumarin without P407 (chemical structure shown in Supplementary Fig. 2) produced local anesthesia slightly longer than from tetracaine (p < 0.05), possibly owing to the large hydrophobic addition to the tetracaine. This indicates that conjugation of the polymer is necessary to inactivate the drug. The duration of local anesthesia bore an almost linear relationship to the irradiation energy density (the product of irradiance and irradiation duration; Fig. 5c), suggesting that the degree of anesthesia could be modulated by varying the irradiance and/or irradiation duration according to patients' changing needs. It would be important for a light-triggerable local anesthetic system to be triggered repeatedly, so as to be able to treat pain over an extended period. To assess the ability of P407-CM-T to provide repeated on-demand local anesthesia, animals were injected in the footpad with 0.1 mL of 20 wt% P407-CM-T (Fig. 5d). No nerve block ensued. The animals were then irradiated at the site of injection with 400 nm light at 300 mW cm$^{-2}$ for 2 min, resulting in block. Subsequently, irradiation was repeated five times, 15 min after the anesthetic effect from the preceding irradiation event wore off. Each irradiation event resulted in block. The duration of block generally decreased with successive triggering events (Fig. 5e). This was possibly because of drug depletion after each triggering event. Triggering could also be delayed for 2 h or 6 h after injection (Supplementary Fig. 15 and Supplementary Table 1); there was no nerve block until irradiation occurred. Delays in triggering led to fewer triggerable peaks, presumably because of polymer depletion and loss of triggerable material.

**Table 1 Duration of block for different compounds without or with a single irradiation event.**

| Compound | Irradiance (W cm$^{-2}$) | Irradiation duration (min) | Duration of block (min)[a] |
|---|---|---|---|
| Tetracaine | 0 | 0 | 28.6 ± 7.1 |
| CM-T | 0 | 0 | 43.9 ± 11.4 |
| P407-CM-T | 0 | 0 | 0 |
| P407-CM-T | 0.2 | 2 | 19.5 ± 4.5 |
| P407-CM-T | 0.3 | 2 | 36.5 ± 8.9 |
| P407-CM-T | 0.3 | 5 | 78.1 ± 24 |
| Saline | 0.3 | 2 | 0 |

[a]mean ± SD.

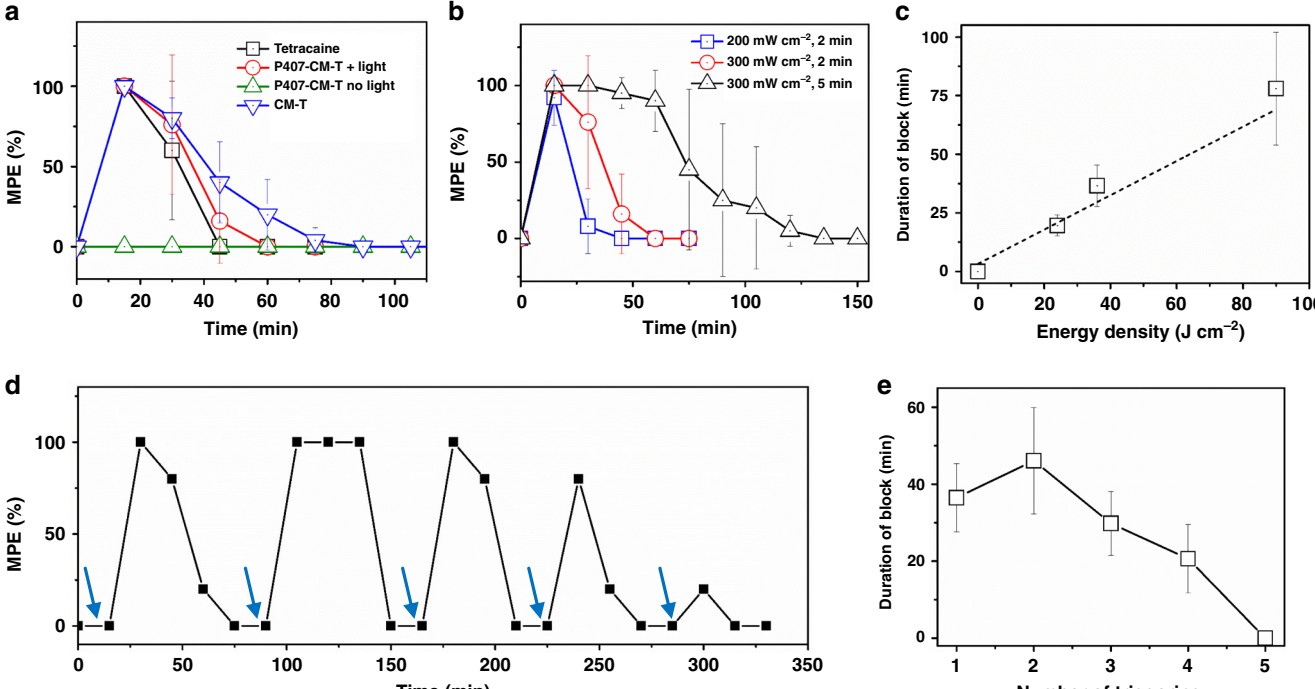

**Fig. 5 Photo-triggered local anesthesia in the rat footpad. a** Time courses of nerve block after injection of tetracaine (black line, n = 6), CM-T (tetracaine bound to coumarin without P407, blue line, n = 5) and P407-CM-T with (red line, n = 4) or without (olive line, n = 4) 2 min irradiation at 300 mW cm$^{-2}$, immediately after injection. **b** Nerve block after irradiation of P407-CM-T with irradiation at various irradiances and durations (blue line: 200 mW cm$^{-2}$, 2 min; red line: 300 mW cm$^{-2}$, 2 min; black line: 300 mW cm$^{-2}$, 5 min; n = 4). (The red plot is also in **a**) **c** Effect of energy density on the duration of the triggered nerve block (n = 4). **d** Representative time course of nerve block with multiple light triggering events (blue arrows represent LED triggering for 2 min at 300 mW cm$^{-2}$). **e** The mean duration of block after each triggering event in **d** (n = 4). Data are means ± SD. Source data are provided as a Source Data file.

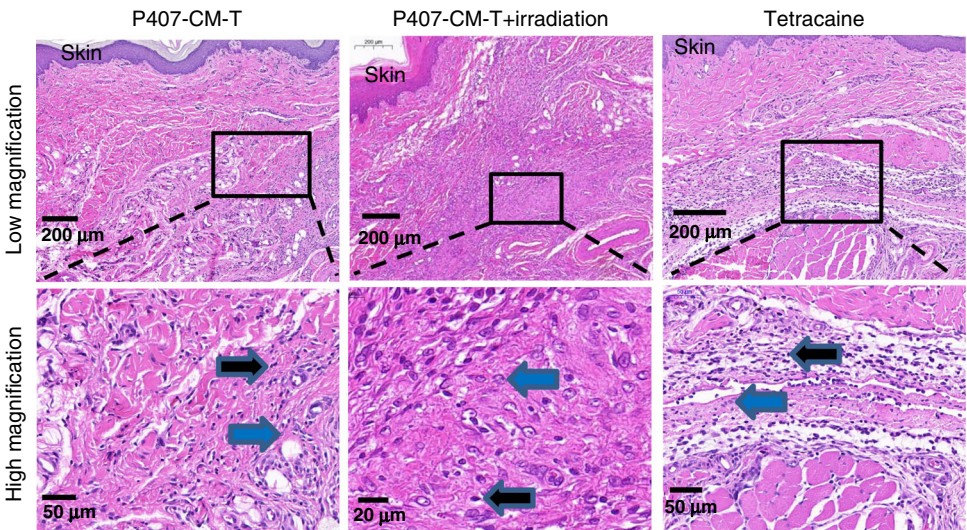

**Fig. 6 Tissue reaction to rat footpad injections on day 4 after injection of P407-CM-T (with/without irradiation) or tetracaine.** Inflammation at the injection site was characterized by lymphocytes (black arrows) and macrophages (blue arrows) extending from the sub-epidermal layers into the deeper muscular layer. Panels on bottom (scale bar: 50 μm) are magnified views of the outlined sections in the panels on the top (scale bar: 200 μm). Data are representative of four animals in each group.

Tetracaine, like all amino-ester and amino-amide local anesthetics, can cause systemic toxicity (e.g., cardiac arrhythmias, seizures) when given in excessive doses or in cases of inadvertent intravascular injection. None of the animals tested had evidence of nerve block in the uninjected (contralateral) extremity; the latter is a useful metric of systemic drug distribution[37,38]. All of the animals were well-appearing, and none developed respiratory distress, had seizures, or died.

**Tissue reaction**. To assess tissue reaction to the formulations, rats were killed 4 days and 14 days after injections, and their foot pads were harvested for histological analysis (Fig. 6 and Supplementary Figs. 16–21). Local anesthetics, in solution[39,40] or in sustained release systems[41,42], are known to potentially cause inflammation, myotoxicity, and neurotoxicity. The last two are not seen well in this subcutaneous model. There was no evidence of tissue (cell) injury in any group. Inflammation in all groups was mild to moderate, and consistent with what is commonly seen after injection of biomaterials and/or local anesthetics[41,43,44]. Inflammation consisted of lymphocytes and macrophages, and generally was diminished by day 14 after injection. Irradiation itself did not cause inflammation (Supplementary Fig. 21).

## Discussion
Triggered local anesthesia would allow patients to adjust their pain relief as needed without needing systemic medications such as opioids or procedures requiring skilled personnel (at least after the initial application). Light, owing to its tunable wavelength, irradiance, area, and duration of exposure, is a suitable trigger. Laser sources are commonly used in light-triggered drug-delivery systems owing to their monochromaticity, coherence, and high intensity. However, laser sources can be expensive and relatively bulky, and their high intensity can cause tissue injury. Light from LEDs is not monochromatic or coherent, and is generally comparatively low in intensity. Also, LEDs are cheaper and can be small and inexpensive. The light source in the current system was a blue light LED which could potentially be suitable for real-world applications.

400 nm light is not expected to penetrate deeply into tissue. However, we have demonstrated that the light that penetrated through the skin was sufficient to cleave the bonds between tetracaine and coumarin and release the free drug. This specific formulation could be applicable in situations where the nerves of interest are relatively close to the body surface, e.g., dental applications.

We have developed a macromolecular prodrug of tetracaine that does not cause local anesthesia in the absence of photo-triggering, and provides LED-triggered local anesthesia in proportion to the intensity and duration of irradiation. This general approach could be extended to triggered release of other compounds.

## Methods
**Synthesis**. Reagents, synthetic routes (Supplementary Figs. 1, 2) and synthetic procedures are described in detail in Supplementary Information.

**Instruments and characterization of materials**. $^1$H and $^{13}$C NMR experiments were measured on a Varian 400 M or 500 NMR spectrometer. The spectra were referenced to the residual solvent peak in CDCl$_3$ at δ 7.27 ppm for $^1$H proton and δ 77.00 ppm for $^{13}$C, respectively. An Agilent 1260 series high performance liquid chromatography (HPLC) with a UV-vis detector was used for analyzing the released drug in the releasing experiments. The mobile phase was 40/60 acetonitrile/water with flow rate of 0.5 mL min$^{-1}$. An Agilent 1200 Series LC-MS with a 6130 Quadrupole MS detector was used for the LC-MS experiments. GPC were measured in THF at 35 °C on Tosoh EcoSEC instrument. The flow rate was 0.35 mL min$^{-1}$. UV-vis absorption spectra were measured on an Agilent 8453 UV-vis G1103A spectrometry. Photo-triggering experiments were performed using a BLS-13000-1 LED driver and an LCS-0405-50-XX lamp from MIGHTEX. Rheology experiments were tested on a TA DHR-2 rheometer with a frequency of 1 Hz, shear strain of 1% and a temperature ramp between 10 °C and 45 °C.

**Sample preparation for injection**. Tetracaine hydrochloride, P407, or P407-CM-T were dissolved in water. CM-T was dissolved in a small amount of ethanol, and an equimolar amount of HCl (2 M in ethanol) was added. The solvent was completely removed by rotatory evaporation, and the resulting CM-T hydrochloride salt was re-dissolved in water.

**Photocleavage of P407-CM-T in vitro**. To demonstrate the capability of P407-CM-T to release tetracaine in the native form, 100 μL of P407-CM-T solution (10 mg mL$^{-1}$) was placed in a 2 mL centrifuge tube and irradiated with 400 nm LED light at irradiances and for durations described in the body of the paper. The solution was then diluted to 1 mL to create volume for subsequent measurements. HPLC or LC-MS was used to verify the release of tetracaine and determine its concentration. Dark stability of P407-CM-T was evaluated by placing the solution of P407-CM-T in a 37 °C oven, followed by HPLC measurements at predetermined intervals.

**Cytotoxicity analysis.** C2C12 and PC12 cells were used to assess cytotoxicity of the materials to muscle and nerve cells respectively. Different compounds were directly added into the cell culture media (1 mg mL$^{-1}$, 0.33 mg mL$^{-1}$, and 0.11 mg mL$^{-1}$) and incubated in the media bathing the cells (i.e., in direct contact with them) in conventional cell culture wells (Supplementary Fig. 3a). After 24 h, cell viabilities were evaluated with the MTS assay, and their survival expressed as percentages of results in untreated cells.

Owing to the high molecular weight of the polymer-containing samples, their molar concentrations were perforce lower than that of the small molecule tetracaine. It would not be practical to culture cells in media containing the polymer concentration to be used in vivo (20 wt%), which would be very viscous. Therefore, 0.05 mL of test materials (0.5 wt% tetracaine or 20 wt% P407 or 20 wt% P407-CM-T) were placed in Transwell inserts such that they were in continuity with the cell culture media (Supplementary Fig. 3b). After 24 h, cell viabilities were evaluated with the MTS assay, and their survival expressed as percentages of results in untreated cells.

**Animal studies.** Animal studies were performed according to protocols approved by the Boston Children's Hospital Animal Care and Use Committee. Adult Sprague-Dawley rats (300–400 g, from Charles River Laboratories) were housed in groups under a 12 h/12 h light/dark cycle. Under brief isoflurane–oxygen anesthesia, 100 μL of solution were injected into the plantar aspect of the rat hind paw. Neurobehavioral testing was performed at predetermined intervals by stimulating the rat footpad with Touch Test sensory evaluators (filament with target force of 180 g) and noting the vocal or motor response (foot withdrawal) of the rat[17,30], as modified from previous reports[31–34]. No vocalize or withdraw after five trials were defined as complete nerve block or 100% MPE. The duration of nerve block was calculated as the time that nerve block was >50% MPE. The baseline pain level of animals was assessed by testing with filaments with gradually increased target forces (26 g, 60 g, 100 g, 180 g). Uninjected animals responded to forces of 60 g or higher. The force required to elicit a response did not change after injection of 100 μL of saline or 20 wt% P407-CM-T without subsequent irradiation. In photo-triggered anesthesia experiments, the hind paw of the rat with injected formulations were irradiated by the LED at designed intensity for 2–5 min under brief isoflurane–oxygen anesthesia. The diameter of the LED light source was 2.5 cm, which can cover the entire footpad.

**Statistics.** Data were described with means and standard deviations (SD) calculated from Microsoft excel (365ProPlus). $F$ test was used to determine whether two samples have different variance, then unpaired $t$ test was used to calculate the $p$ values from the software of Microsoft excel. Measurements were taken from distinct samples. OriginPro 9.1 was used for plotting.

**Histology.** The rats were killed with carbon dioxide 4 days or 14 days after injection. The rat foot pads were dissected, fixed in formalin, embedded in paraffin and underwent standard processing to produce hematoxylin and eosin-stained slides.

**Reporting summary.** Further information on research design is available in the Nature Research Reporting Summary linked to this article.

## Data availability

The data in this work are available within the paper and its Supplementary Information files.

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

## Acknowledgements

This study was funded by National Institutes of Health (NIH) grant R35GM131728.

## Author contributions

W.Z. and D.S.K. designed the experiments. W.Z., T.J., Y.L., Y.Z., C.Z., and A.L. performed the synthesis, material characterizations, and in vitro studies of the materials. W.Z., T.J., and Y.L. performed the animal experiments. M.M. analyzed the histology. W.Z. and D.S.K. analyzed the data and wrote the paper.

## Competing interests

The authors declare no competing interests.
