## [Peer Review File · Nature Communications]

Reviewers' comments:

Reviewer #1 (Remarks to the Author):

This is a proof of concept study demonstrating the potential of using light as a cue for the on-demand release of local anesthetics. The manuscript is well written and the provided data support the conclusions. Below are the comments.

Introduction

Clearly outlined the background and the rationale for the study

Results and Discussion

Data clearly showed the light triggered release of drug from the polymer conjugated form and its in vivo efficacy.

The inflammation in response to the drug conjugated polymer injection raises some concerns. It is not clear whether injection of the polymer conjugates raises the base pain level compared to the control group. Please include a discussion.

Also, it is not clear if the approach presents limitations in terms of how much drug can be injected at a time.

Reviewer #2 (Remarks to the Author):

The article presents the development and pharmacological evaluation of a new local anesthetic delivery system triggered by light. Then, it is an important contribution for possible novel on-demand controlled analgesia systems. Also, is an innovative idea and experiments are well-designed. However, some points need to be clarified:

General comments:

- 1- What are the clinical vantages of the developed system compared to other stimuli responsive and on-demand available devices?
- 2- Why tetracaine was chosen as drug model? What is the clinical relevance of tetracaine for post-operative pain management (considering this case as a possible clinical application)?
- 3- The authors proposed clinical applications for dental anesthesia (gingival and periodontal pocket injection and/or infiltrative anesthesia?), but the formulation was evaluated just by infiltrative anesthesia in soft tissues. How do authors can discuss this possible application considering vascular aspects (also the drug uptake to systemic blood stream) and tissues morphological differences?
- 4- What about the tetracaine systemic side-effects? Were those effects evaluated?

Specific comments:

- 1- Figure 3c: two irradiation potencies were tested, and 50 mW/cm² irradiation induces almost 100 % of cleavage at 2 min. Are tetracaine concentrations sufficiently released in this condition?
- 2- The cleavage percentage was evaluated considering 25 and 50 mW/cm². However, for in vivo tests, 200 mW or 300 mW/cm² were used. How those potencies were selected?
- 3- What was the real irradiation area during the in vivo assays? If this is a depot formulation (considering the P407 features and concentration used), how the irradiated area was standardized?
- 4- Lines 180 to 182: in vitro assays showed a pronounced cell viability reduction after treatment with tetracaine, but this effect was not observed for P407-CM-T. Even considering the differences between in vitro and in vivo conditions, how authors explain the local toxic effects induced by P407-CM-T? If the local tissue reaction was similar to that observed for tetracaine injection, what is the vantage of using this system regarding to local toxic effects?

Response Letter

Thank you for handling our manuscript. We appreciate the reviewers' positive and insightful comments. Our point-by-point responses are provided below in **bold**:

Reviewer #1:

(1) This is a proof of concept study demonstrating the potential of using light as a cue for the on-demand release of local anesthetics. The manuscript is well written and the provided data support the conclusions.

Reply: **We thank the reviewer for these positive comments.**

Below are the comments.

Introduction

Clearly outlined the background and the rationale for the study

Results and Discussion

Data clearly showed the light triggered release of drug from the polymer conjugated form and its in vivo efficacy.

Reply: **We thank the reviewer for these positive comments.**

(2) The inflammation in response to the drug conjugated polymer injection raises some concerns.

Reply: **Inflammation is virtually ubiquitous with injected materials, occurring even with saline (as we show in figure S20), and certainly with injected local anesthetics – free or in sustained release systems – including in all local anesthetic formulations in current clinical use. In response to these points, we have rewritten the relevant section at Line 178:**

“To assess tissue reaction to the formulations, rats were euthanized 4 days and 14 days after injections, and their foot pads were harvested for histological analysis (Figure 6 and S16-21). Local anesthetics, in solution^{39,40} or in sustained release systems,^{41,42} are known to potentially cause inflammation, myotoxicity, and neurotoxicity. The last two are not seen well in this subcutaneous model. There was no evidence of tissue (cell) injury in any group. Inflammation in all groups was mild to moderate, and consistent with what is commonly seen after injection of biomaterials and/or local anesthetics.^{41,43,44} Inflammation consisted of lymphocytes and macrophages, and generally was diminished by day 14 after injection. Irradiation itself did not cause inflammation (Figure S21).”

It is not clear whether injection of the polymer conjugates raises the base pain level compared to the control group. Please include a discussion.

Reply: **Injection of materials did not affect the baseline pain level. Animals were tested before and after injection. There was no difference before and after injection.**

We added the following to address this in Results and Discussion, line 143:

“Injection of 100 uL of saline or P407-CM-T (20 wt%) did not affect response to the filament (Figure S14).”

And in SI at Line 228”

“The baseline pain level of animals was assessed by testing with filaments with gradually increased target forces (26g, 60g, 100g, 180g). Uninjected animals responded to forces of 60g or higher. The force required to elicit a response did not change after injection of 100 uL of saline or 20 wt% P407-CM-T without subsequent irradiation.”

Figure S14. Percentage of animals (n = 5) that responding to filaments.

It also bears mentioning that injection of drug delivery systems (in the absence of free drug) does not cause nerve block. We have shown this with dozens of drug delivery systems.

(3) Also, it is not clear if the approach presents limitations in terms of how much drug can be injected at a time.

Reply: **The volume selected here was limited to 100 µL due to the size of the rat foot pad. This does not mean that there is an inherent limitation on volume or dose to be injected, except to the extent that all peripheral nerve blocks, even in humans, have a volume limitation. (For example, one typically injects 25-30 mL at the human sciatic nerve; that is not a limitation of clinical effectiveness.)**

Reviewer #2:

The article presents the development and pharmacological evaluation of a new local anesthetic delivery system triggered by light. Then, it is an important contribution for possible novel on-demand controlled analgesia systems. Also, is an innovative idea and experiments are well-designed.

Reply: **We thank the reviewer for the positive comments.**

However, some points need to be clarified.

General comments:

1- What are the clinical vantages of the developed system compared to other stimuli responsive and on-demand available devices?

Reply: **As we noted in the introduction “[Previous triggered local anesthetic sustained release systems] suffered from a problem common to most particulate drug delivery systems: release occurring from the moment of the devices’ creation until drug is depleted. Early on, this results in untriggered rapid drug release; in the context of local anesthesia, this may result in extended initial nerve block, which may be undesirable. Subsequently, ongoing release may result in depletion of drug even if the system is not triggered (i.e. basal release), so that it is no longer available for triggered release. To eliminate the unwanted initial blockade and basal drug release, we conjugated the drug onto macromolecular carriers in a manner that could be reversed by photo-triggering.”**

This distinction (lack of basal drug release) differentiates our work from most sustained release systems – triggered or not – for local anesthetics and other compounds. As noted in the introduction, this difference prevents unwanted nerve blockade and depletion of drug.

A minor difference (from the scientific but not the practical point of view) between this work and our previous work with triggered local anesthetics is that here were used the “conventional” amino-ester agent tetracaine, which is in widespread clinical use, rather than tetrodotxin, which is still a largely experimental agent.

2- Why tetracaine was chosen as drug model? What is the clinical relevance of tetracaine for post-operative pain management (considering this case as a possible clinical application)?

Reply: **Tetracaine was chosen because (i) it is a conventional local anesthetic in widespread clinical use. (ii) It has relatively high potency in the class of conventional local anesthetics. (iii) It has a secondary amine group that could be conveniently modified.**

We added a short supplement about this at Line 57:

“Tetracaine was selected as the local anesthetic in this work due to its widespread clinical use, relatively high potency in its class and the presence of a modifiable secondary amine group”

3- The authors proposed clinical applications for dental anesthesia (gingival and periodontal pocket injection and/or infiltrative anesthesia?), but the formulation was evaluated just by infiltrative anesthesia in soft tissues. How do authors can discuss this possible application considering vascular aspects (also the drug uptake to systemic blood stream) and tissues morphological differences?

Reply: We used an infiltrative model of anesthesia, but it is well-known that tetracaine – like all amino-amide and amino-ester local anesthetics, can be used for infiltration, peripheral nerve block, topical anesthesia, etc. As we have shown in other work, effectiveness in infiltration anesthesia generally predicts effectiveness in other types of nerve block, although not necessarily in a 1:1 ratio. There is no obvious reason why the present formulation would not be usable to block, for example, the superior alveolar nerve, provided that the nerve was at a tissue depth that could be reached at a safe irradiance (which it should be). In general, local anesthetics that can be used in one anatomic location can be used in others; differences in local blood flow etc. are not generally primary determinants of their clinical use. (This is not to say that clinical effect does not vary between anatomic locations, but it is often due to factors such as the maximum volume that can be injected, not properties of the drug.)

The limitations of this system, as far as the local anesthetic itself is concerned, are generic to all local anesthetics and are not particular to the triggering mechanism or the specific drug.

4- What about the tetracaine systemic side-effects? Were those effects evaluated?

Reply: The principal side-effect of amino-ester local anesthetics in this model is likely to be local (tissue toxicity) which was evaluated by histology. We have made this more clear as follows at Line 178:

“To assess tissue reaction to the formulations, rats were euthanized 4 days and 14 days after injections, and their foot pads were harvested for histological analysis (Figure 6 and S16-21). Local anesthetics, in solution^{39,40} or in sustained release systems,^{41,42} are known to potentially cause inflammation, myotoxicity, and neurotoxicity. The last two are not seen well in this subcutaneous model. There was no evidence of tissue (cell) injury in any group. Inflammation in all groups was mild to moderate, and consistent with what is commonly seen after injection of biomaterials and/or local anesthetics.^{41,43,44} Inflammation consisted of lymphocytes and macrophages, and generally was diminished by day 14 after injection.”

Amino-amide and amino-ester local anesthetics can also cause systemic toxicity. However, that is extremely rare in this model at the doses used. The absence of systemic toxicity is seen in the absence of deficits in the uninjected extremity, a reliable marker of systemic toxicity, or any other signs of toxicity.

To address this, we have added the following to the manuscript at Line 172:

“Tetracaine, like all amino-ester and amino-amide local anesthetics, can cause systemic toxicity (e.g. cardiac arrhythmias, seizures) when given in excessive doses or in cases of inadvertent intravascular injection. None of the animals tested had evidence of nerve block in the uninjected (contralateral) extremity; the latter is a useful metric of systemic drug distribution.^{37, 38} All of the animals were well-appearing, and none developed respiratory distress, had seizures, or died.”

Specific comments:

1- Figure 3c: two irradiation potencies were tested, and 50 mW/cm² irradiation induces almost 100 % of cleavage at 2 min. Are tetracaine concentrations sufficiently released in this condition?

Reply: The purpose Figure 3 was to demonstrate that the drug can be cleaved from the polymer in its native form *in vitro*, and as a secondary point that the degree of cleavage depends on the irradiance. The sufficiency of those concentrations for effectiveness *in vivo* is difficult to determine without doing the actual experiment (see response to next comment), hence Figure 5.

2- The cleavage percentage was evaluated considering 25 and 50 mW/cm². However, for *in vivo* tests, 200 mW or 300 mW/cm² were used. How those potencies were selected?

Reply: As we have demonstrated elsewhere (*J. Control. Release* 2018, 286, 55-63.), the irradiance needed to phototrigger a drug delivery system depends on many factors, including the depth and type of intervening tissue, which would attenuate the light. Thus, it was inevitable that triggering *in vivo* would require a greater irradiance. The specific irradiances used to provide nerve block within a brief irradiation time without causing thermal injury were based on pilot studies, the tail end of which is shown here (the escalation from 200 to 300 mW/cm²). When 300 mW/cm² was used for the same *in vitro* testing, complete cleavage occurred within one minute. Data were added to Figure 3c (blue curve):

We have added commentary on this at Line 147:

“(Higher irradiances were used *in vivo* compared to *in vitro* due to attenuation of light by traversing tissues.³⁶)”

3- What was the real irradiation area during the *in vivo* assays? If this is a depot formulation (considering the P407 features and concentration used), how the irradiated area was standardized?

Reply: **The area of the whole foot pad, which was about 1 cm wide, was irradiated since the diameter of the LED light source was 2.5 cm.**

We added a description at Line 234 in SI:

“The diameter of the LED light source was 2.5 cm, which can cover the entire foot pad.”

4- Lines 180 to 182: *in vitro* assays showed a pronounced cell viability reduction after treatment with tetracaine, but this effect was not observed for P407-CM-T. Even considering the differences between *in vitro* and *in vivo* conditions, how authors explain the local toxic effects induced by P407-CM-T? If the local tissue reaction was similar to that observed for tetracaine injection, what is the vantage of using this system regarding to local toxic effects?

Reply: **There are a number of related points here.**

1) It is important to distinguish between inflammation and cytotoxicity. What was observed *in vivo* was inflammation, which is not the *in vivo* correlate of the cytotoxicity observed *in vitro* – which would be actual cell injury. In this *in vivo* model, the injection was subcutaneous, and the histological samples did not include much muscle or visible nerve fibers, making it difficult to assess myotoxicity or neurotoxicity. (Those would have been the toxicities to compare most directly to the *in vitro* findings.)

2) The inflammation seen with P407-CM-T was not surprising. Inflammation is almost uniformly induced by injection of foreign materials, including hydrogels, liposomes, polymeric particles, etc. Inflammation can also be caused by local anesthetics. (In fact, as is well known and we show here, injection of saline can do it as well.) Since inflammation was expected, the purpose of the histology was therefore not so much to show differences between groups, but to see whether there was evidence of cellular/tissue injury.

In scrutinizing the histology again while considering the reviewer's comments, it is clear that inflammation in the various groups (with the exception of the saline group) was all in the range mild to moderate, and actually mild in comparison to what we have seen with some other drug delivery systems (e.g. cross-linked chitosan, poly(lactic co-glycolic acid) microspheres). We have revised this section of the manuscript to make these points clearer.

In response to these points, we have rewritten the relevant section:

“To assess tissue reaction to the formulations, rats were euthanized 4 days and 14 days after injections, and their foot pads were harvested for histological analysis (Figure 6 and S16-21). Local anesthetics, in solution^{39,40} or in sustained release systems,^{41,42} are known to potentially cause inflammation, myotoxicity, and neurotoxicity. The last two are not seen well in this subcutaneous model. There was no evidence of tissue (cell) injury in any group. Inflammation in all groups was mild to moderate, and consistent with what is commonly seen after injection of biomaterials and/or local anesthetics.^{41,43,44} Inflammation consisted of lymphocytes and macrophages, and generally was diminished by day 14 after injection. Irradiation itself did not cause inflammation (Figure S21).”

We also added Figure S21 in SI:

Figure S21. Tissue reaction 4 days after irradiation (300 mW/cm^2 , 2 min) without injection of test materials. Panel on right is a magnified view of the outlined section in the panel on the left.

Reviewer #2 (Remarks to the Author):

The article presents na innovative contribution for the development of new drug-delivery systems for local anesthesia. All suggestions were addressed and answered by authors.

Response Letter

Thank you for handling our manuscript. We appreciate the reviewers' positive comments. Our point-by-point response is provided below:

REVIEWERS' COMMENTS:

Reviewer #2 (Remarks to the Author):

The article presents na innovative contribution for the development of new drug-delivery systems for local anasthesia. All suggestions were addressed and answered by authors.

Reply: **We thank the reviewer for the positive comments.**